# Evidence Supports PA Prescription for Parkinson’s Disease: Motor Symptoms and Non-Motor Features: A Scoping Review

**DOI:** 10.3390/ijerph17082894

**Published:** 2020-04-22

**Authors:** Yi-Chen Cheng, Chun-Hsien Su

**Affiliations:** 1Department of Exercise and Health Promotion, Colleage of Education, Chinese Culture University, Taipei City 11114, Taiwan; cyz29@ulive.pccu.edu.tw; 2Graduate Institude of Sport Coaching Science, Colleage of Education, Chinese Culture University, Taipei City 11114, Taiwan

**Keywords:** Parkinson’s disease, Parkinson’s disease dementia, physical activity, motor disorders, quality of life, prescription

## Abstract

Parkinson’s disease (PD) is a prevalent neurodegenerative disorder, which relates to not only motor symptoms, but also cognitive, autonomic, and mood impairments. The literature suggests that pharmacological or surgical treatment has a limited effect on providing relief of the symptoms and also restricting its progression. Recently, research on non-pharmacological interventions for people living with PD (pwPD) that alleviate their motor and non-motor features has shown a new aspect in treating this complex disease. Numerous studies are supporting exercise intervention as being effective in both motor and non-motor facets of PD, such as physical functioning, strength, balance, gait speed, and cognitive impairment. Via the lens of the physical profession, this paper strives to provide another perspective for PD treatment by presenting exercise modes categorized by motor and non-motor PD symptoms, along with its effects and mechanisms. Acknowledging that there is no “one size fits all” exercise prescription for such a variable and progressive disease, this review is to outline tailored physical activities as a credible approach in treating pwPD, conceivably enhancing overall physical capacity, ameliorating the symptoms, reducing the risk of falls and injuries, and, eventually, elevating the quality of life. It also provides references and practical prescription applications for the clinician.

## 1. Introduction

Parkinson’s disease (PD) was first defined in 1817 by British doctor James Parkinson based on specific symptoms of his patients, and the name “Parkinson’s disease” has been adopted and used by the public since then. It is a chronic neurodegenerative disorder affecting approximately ten million people worldwide, according to the Parkinson’s Disease Foundation [1]. Its prevalence is estimated at 2802 per 100,000 persons in North America, Europe, and Australia [2], and most of the patients are middle-aged and older populations. Furthermore, from a recent global, regional, and national study, 6.1 million people suffered from PD globally in 2016, which is more than double when compared with 1990 (2.5 million) [3]. The largest increase in the age-adjusted prevalence rates of PD worldwide between 1990 and 2016 was in China, which had increased more than double [3]. Smoking, caffeine intake, and pesticide exposure are well-established risk factors across regions [4].

## 2. PD Symptoms

The symptoms of PD usually develop gradually and are mild at first. There are many different symptoms associated with PD. In general, PD symptoms are characterized by motor and non-motor, and these symptoms gradually deteriorate as PD progresses with time [5]. There are four primary motor symptoms of Parkinson’s disease: tremor, rigidity, bradykinesia (slow movement), and postural instability (balance problems) [6]. In terms of non-motor features, most of the people living with PD (pwPD) experience symptoms such as disturbances in the sense of smell, sleep problems, depression and anxiety, pain, psychosis, fatigue, cognitive changes, weight loss, gastrointestinal issues, lightheadedness, urinary issues, sexual concerns, sweating, melanoma, personality changes, and eye and vision issues. However, the order in which these develop and their severity is different for each individual.

There is high variability in PD’s disease progression and symptoms. With each pwPD’s different physical and mental state, PD’s impact may differ from each pwPD, such as functional capacities, quality of life, and social participation. The key factors that affect pwPD’s quality of life are depression, physical disorder, the severity of the disease, insomnia, pain, and cognitive impairment.

## 3. Benefits of Physical Activity

Besides pharmaceutical treatment, physical activity (PA) can be seen as a complement to manage the indigenous decline associated with the disease. This integration with PA was introduced during the 1950s. PA was considered as a method of minimizing the limitation caused by PD. Many researchers favored the integration of exercises as an essential component of therapy, even with the introduction of levodopa, which had a massive effect on the treatment of PD. The first experimental study was in the 1980s, and even until today, the scientific community recognizes the positive effects of PA on people living with PD. A prospective cohort study clearly reveals the importance of being physically active. Individuals who exercised for 15 min a day or 90 min a week had a 14% reduced risk of all-cause mortality and had a 3-year longer life expectancy [7]. This result shows great influences by PA, and the benefits are applicable to all age groups and both sexes; thus, this paper regards applying PA as the significant treatment for PD symptoms since medication aids have limited effects, let alone undesired side effects. Exercise is important throughout all symptom progression phases of PD, starting from early symptoms, motor fluctuations, dyskinesias, emerging medication-resistant symptoms, even till the last phase of disabling medication-resistant symptoms [8].

According to past reviews, PA seems to enhance motor skills, gait, balance, and muscle strength of pwPD. However, it is still unclear that if PA has positive effects on certain health facets, including superior cognitive functions, activities of daily living, and the psychosocial aspects of life. This review would like to provide factual support from past studies to enhance the promotion and adoption of PA interventions with regards to pwPD’s treatment. Most of the common symptoms with associated PA interventions will be covered in this review.

Subsequently, the overall goal of this paper is to present an overview of the effects of PA on pwPD and to provide a systematic tool as a reference for researchers and clinicians who would consider PA as a therapeutic program for pwPD through collecting and summarizing the health improvements resulting from PA intervention.

Lack of activity destroys the good condition of every human being, while movement and methodical physical exercise save it and preserve it—Plato [9].

The “Exercise is Medicine^®^” (EIM) program was initiated by the American Medical Association by virtue of the thorough studied, documented, and extensively recognized positive effects exercise has on the cardiovascular, metabolic and musculoskeletal systems [10]. Brain health, which relates to blood flow, trophic factors, and immune system changes, particularly gain benefits from exercise, which creates an ideal environment for neuroplasticity [11]. Neuroplasticity infers to a variety of structural and physiological mechanisms such as synaptogenesis, neurogenesis, neuronal sprouting, and potentiating synaptic strength, which leads to the strengthening, repair, and formation of neuronal circuitry, particularly favorable for driving improvements in the injured brain [12].

The American Academy of Neurology and the joint task force of the European Federation of Neurological Societies and European Movement Disorders Society have done literature reviews, suggesting that exercise-based physiotherapy is a catalyst for positive neuroplasticity, and this relates to symptoms of people diagnosed with PD. Exercise improves symptoms in movement, functional capacity, and cognitive function [12]. This is especially effective for those with mild to moderate stage pwPD [10]. However, studies suggest that people diagnosed with PD are still likely to reduce their levels of physical activity [10].

## 4. Types of Exercise

An exercise prescription can appear in numerous forms, but they all follow the FITT principles [13]. FITT stands for frequency, intensity, time (duration), and type (mode). Table 1 below elaborates on the key components and compositions in an exercise prescription. The frequency can refer to the number of exercise sessions per day or the number of exercise sessions per week. Intensity is the amount of effort the person exerts, measured as a percentage. Time is the length of each exercise session and is calculated in minutes. Type refers to which exercise.

According to the United States National Institutes of Health [1], physical exercises are generally grouped into four main types—aerobic exercise, anaerobic exercise, balance and coordination, and flexibility exercise—according to the overall effect on the human body.

### 4.1. Aerobic Exercise

The American College of Sports Medicine (ACSM) defines aerobic exercise as any activity that engages large muscle groups to work continuously and rhythmically in nature [14]. Skills are generally less involved in performing an aerobic exercise. Accordingly, aerobic exercises are usually recommended for all adults to enhance fitness and, with little alternation, can be adapted to varying levels of personal physical fitness such as slow dancing, swimming, leisure cycling, brisk walking, and aqua-aerobics. As a consequence of the aerobic exercise’s representative high intensity, it is recommended for people with regular exercise such as aerobics, jogging, running, stepping exercise, fast dancing, and elliptical exercise.

### 4.2. Anaerobic Exercise

Anaerobic exercise has been defined by the ACSM as intense physical activity of very short duration, and the energy sources are fueled within the contracting muscles [13]. Anaerobic exercises, also known as muscle-strengthening exercises, are activities that enable muscles to work harder than original habituation. A moderate to high level of intensity is involved in performing a muscle-strengthening exercise. The major muscle groups, including the shoulders, arms, chest, abdomen, legs, hips, and back, are required during muscle-strengthening exercises. Resistance training is one of the popular examples of muscle-strengthening exercise.

### 4.3. Flexibility Exercise

Flexibility, defined by the ACSM, refers to the range of motion available at a joint [15]. For this reason, flexibility activities are a reasonable part of an exercise program, even though they have not known health benefits and it is unclear whether they reduce the risk of injury [16]. Muscle stiffness is typical of people living with PD. In consequence, joint and muscle flexibility is important, and stretching should be the first step in the exercise program. Stretching helps fight the muscle rigidity that comes with PD, especially make pwPD’s daily tasks and movements easier.

### 4.4. Balance And Coordination

According to ACSM’s definition, balance is the maintenance of equilibrium while static or mobility [15]. Balance is regarded as the ability to maintain upright or staying in control of body movement, and coordination is the ability to control body movements with more than two body parts smoothly and efficiently. In addition to good balance, good levels of other fitness aspects like strength and agility are also required in managing the complex skill, coordination. With the help of practice and training from particular sports, balance and coordination are improvable.

## 5. Exercise Prescriptions for Treating PD Symptoms

Each exercise type helps train different muscle groups, so, for pwPD, integrated training is the key prescription. One shall not favor any particular type and neglect the interlocking relationships between each exercise type.

Figure 1 aggregates the time and degree of disability pwPD suffer; along the way, this paper suggests compatible recommended exercise to alleviate each motor or non-motor features. There are four different stages shown in Figure 1. Non-motor features occur in the very beginning phase throughout the progress and then the motor symptoms in the prodromal stage. The severity of both motor symptoms and non-motor features shown in the *y*-axis increases as the stage of PD progresses, where motor symptoms appeared as an upward line and non-motor features appeared as a downward line.

Various symptoms across the stages of PD have corresponding exercise suggestions presented in the vertical section of each symptom. For example, pwPD with bradykinesia, rigidity, and tremor can be treated with PA, such as cardiovascular, balance and coordination, and stretching. Significant studies that conducted RCTs on patients with early- and mid-stage PD indicate the benefit of aerobic exercise on reducing PD symptoms and improving cardiovascular fitness and long-term aerobic endurance [17,18]. Consequently, aerobic exercise is shown as being corresponding to early- and mid-stage PD in Figure 1. With the integration of corresponding exercise and original figure from the study, Parkinson disease, from Poewe et al., Figure 1 visually presents a PD symptoms timeline with exercise treatment [19].

### 5.1. Exercise for Motor Symptoms

This paper strives to provide exercise insights for pwPD. Table 2 and Table 3 below indicates the suggested exercise modes for motor and non-motor features of PD.

For targeting PD’s motor symptoms such as freezing of gait, motor skill dysfunction, weak muscles, imbalance, and rigidity, different exercise modes are listed for improvement. A recent study about exercise strategy for motor symptoms of PD, focusing on gait and balance, clearly suggests that there are seven kinds of exercise which may improve gait and balance, including muscle strength training, aerobics, cueing exercise, gait training, balance training, Tai Chi, and dance [20,21].

### 5.2. Cueing Strategies

Even though recent systematic reviews could not gain concrete evidence supporting the efficacy of physiotherapy in PD due to methodological problems in some studies, reviewers did state that additional cueing techniques could improve the efficacy of physiotherapy [39]. Cueing is described as using a piece of information or stimuli to aid or facilitate movement initiation. Recent reviews on cueing reveal that it can have an immediate and supreme effect on gait performance in pwPD, such as improvements in walking speed and steps [40]. For instance, cycling is demonstrated as a useful approach as exercise training in pwPD who are grounded by severe freezing of gait [22]. Moreover, recent studies even observe an improvement of freezing of gait by applying nonexternal cues, which are imagining and mimicking bicycling by pwPD [23].

Previous laboratory experiments have shown the influence of cueing in single-session results in a short-term correction of gait; however, if carry-over to un-cued performance and activities of daily living (ADL), the influence is limited [41]. It becomes very complex when experimenting cues in therapeutic settings, as each individual pwPD has different triggers of cue delivery, such as visual, auditory, or somatosensory, and the cue parameter adopted for movement correction such as frequency or size of step. Despite the fact that there are two studies mentioning the retention effects of cues, the training effects and the clinical application of cues for improving walking and gaits have not been discussed and evaluated in other work. Besides, issues such as being distracted, which may lead to increasing the risk of falling, have risen while applying cues.

External cues are used as effective interventions aiming to improve motor performance, especially gait [24]. In order to improve pwPD’s everyday motor tasks, cognitive movement strategies present mental techniques by teaching the pwPD to separate complex motor sequences in series of single and simple movements that are to be performed in a correct and unchanged order. This strategy aims to offer motor performance as a conscious task, meanwhile avoiding multi-tasking, which also bypasses the defective basal ganglia. Balance training is another pivotal part, which links tightly with fall prevention. Additionally, improving physical capacity with aerobic training, flexibility, and strength exercises may reduce symptoms and improve pwPD’s general well-being and quality of life.

### 5.3. Exercise for Non-Motor Features

Table 3 is a classification by PD’s non-motor features, which provides insight for which exercise can assist with PD non-motor features, along with the improvement effects.

### 5.4. Balance Training

Basal ganglia degeneration involves a number of physiological systems essential for balance control. Dysfunctional basal ganglia would deteriorate the central nervous system’s ability to translate sensory information, including somatosensory, vestibular, and visual, into a single reference frame, which is critical for assessing limb and body positioning with regard to the environment. Motor regulation deficient in PD arises from problems of adopting postural synergies, inadequate inter-segmental coordination, and delayed response in motor commands when working within different tasks. Specifically, it is essential to involve balance training to improve functions, or impairments, of balance control in relation to PD symptoms.

A weak ability to adapt to balance changes causes the increased risk of falling and the resulting injuries, e.g., hip fracture. It is evident that an integrative therapy for improving the response of pwPD is a necessity. Consequently, scholars and clinicians have developed and evaluated exercise interventions, particularly on balance.

Tai Chi, a Chinese martial arts discipline, has been proven as a useful exercise method for pwPD since it incorporates diverse techniques such as weight shifting, slow and controlled movement, trunk rotations, different stances, multidirectional stepping, and maintenance of postures that directly target PD balance and gait [20].

Dancing is another program that is considered effective for treating PD symptoms that are directly associated with dancing movements, including turning, backward walking, dynamic balance, and multitasking skills. On the other hand, the essence of dance includes social and enjoyable facets that could benefit the emotional and psychological aspects of pwPD and further build pwPD’s social network.

Balance impairments are the top concern of all motor impairments affecting pwPD. Balance is highly related to falls risk, thus exercises such as Tai Chi, dance, biking, and boxing, are supported by research showing the positive effects on pwPD due to their focus on training weight shift and postural control.

### 5.5. Exercise for Cognitive Dysfunction

There are laboratory studies articulating the effects of exercise on the brain health of pwPD. Effects include neuro-protection (slow, negate, or reverse the neuro-degenerative process) and neuro-restoration (adaption of compromised neural pathways).

Studies have shown that regular exercise both triggers and maintains the production of glial cell line-derived neurotrophic factor (GDNF) producing cells in the substantial nigra where DA neurons are located, and then lead to an increase in dopamine release [67]. From a pharmacological treatment point of view, the finding is especially significant due to the reduction of relying on levodopa, going forwards for patients with early PD symptoms. In regard to the effectiveness of reducing PD symptoms such as tremor at rest, rigidity, and difficulty in exercising, levodopa is considered the most effective drug that, on the other hand, also causes unwanted effects such as dyskinesias. With the involvement of exercise intervention, these unwanted effects can potentially be minimized or postponed.

Cognitive dysfunction is a severe symptom that pwPD suffer. In clinical practice, cognitive training is usually carried out separately from mobility training. From a former review, this training is effective for people living with PD [25,26].

Recent investigations have demonstrated improvements in the freezing of gait (FOG) after cueing or mobility interventions. However, the setting is done under the integration of cognition and mobility (rather than separately); research to date has not investigated the integration of freezing-specific motor and cognitive therapies for PD people with FOG. Research investigating the effects of targeted cognitive and motor interventions on freezing of gait severity is needed. Targeting Parkinson’s disease motor symptoms, exercise modes and improvements are listed in Table 4, which strive to provide an initial point for evaluation of such targeted cognition–mobility training in pwPD who suffer from motor symptoms.

A recent study showed that general exercise may be as effective as LSVT BIG therapy on pwPD symptoms [72]. LSVT is a speech treatment protocol which stands for Lee Silverman Voice Treatment, and LSVT BIG is considered an intensive, effective, one-on-one treatment created to help people with Parkinson’s disease and other neurological conditions address walking, balance, and other activities of daily living through training people to increase the size (amplitude) of their movements from head to toe like buttoning or writing [73]. BIG siginifies the technique of increasing the size (amplitude) of pwPD’s movements in LSVT BIG therapy.

It is suggested that a combination of skilled and aerobic training is best to trigger and maintain multiple mechanisms of neurogenesis. Forced rate, in other words, beyond what the participant would self-select, is recommended as well. Regarding the early phases of motor development and maximizing motor learning, the prefrontal cortex is a critical part. Cognitive engagement is required through verbal, proprioceptive feedback, dual tasking, cueing, and motivation.

## 6. Conclusions

People living with PD are known to suffer from FOG, limited stride length, and balance impairments. This paper provides tailored physical activity as an intervention to help build pwPD’s daily task abilities and consequently have a better quality of life. Numerous studies have demonstrated the benefits of different types of physical activity. Consequently, providing corresponding exercise intervention for pwPD is an optimal option other than intrusive treatment such as medication and surgery.

Studies have articulated cueing strategies, cognitive movement strategies, balance training and strength, and flexibility training as effective for treating pwPD. Aiming for each PD deficiency, this paper strives to provide different physical training protocol according to specific physical characteristics. By systematizing PD symptoms with motor, non-motor, and cognitive domains, providing each symptom with corresponding exercise interventions can be a useful tool for researchers, clinicians, families, and pwPD.

PD is a very complex disease, and each pwPD suffers from different symptoms even at the same stage, let alone in different stages. Consequently, the adoption of an individualized approach is strongly recommended. Customization of the pwPD’s personal condition targets his or her precise motor impairments. However, this study targets to provide general exercise advice in order to provide pwPD and pwPD’s families a guide to fighting against different impairments.

Subsequently, educating pwPD, their families, and clinicians about the importance of an active lifestyle is crucial. The adoption of regular physical activity in daily lives raises pwPD’s daily routine ability, independence, physical functionality, and, ultimately, the quality of life.

## Figures and Tables

**Figure 1 ijerph-17-02894-f001:**
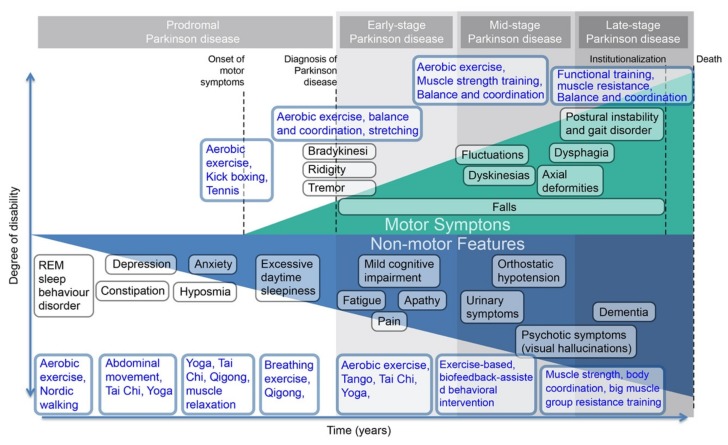
People living with PD (pwPD) symptoms timeline. REM—rapid eye movement.

**Table 1 ijerph-17-02894-t001:** The FITT Principle of Prescribing Aerobic Exercise.

Key Components	Compositions to be Specified in an Exercise Prescription
Frequency (F)	The number of exercise sessions per day or the number of exercise sessions per week
Intensity (I)	The amount of effort the person exerts, measured as a percentage
Time (T)	The length of each exercise session and is calculated in minutes.
Type (T)	The mode of exercise performed.

**Table 2 ijerph-17-02894-t002:** Exercise mode and improvement for motor symptoms of Parkinson’s disease.

Symptom	Exercise Mode	Improvement	References: Grade ^2^
Freezing of gait	Running, Swimming, Cycling, Hiking, Aerobic dancing, Jogging Skipping rope, Running, Cycling, Kick-boxing, Playing tennis Heel-to-toe walk Tai Chi	Walking speed↑ ^1^ Step and stride length↑ Motor performance↑ Cadence↑ Cardiovascular endurance↑ Gait speed↑ Balance↑ Weight shifting ability↑ Movement control↑ Trunk rotation↑ Multidirectional stepping ability↑ Maintenance of postures↑ Walking speed and steps↑	[21]: A, [22]: C, [23]: C, [24]: B, [25]: D, [26]: A
Motor skill dysfunction	Kick-boxing Playing tennis Push-ups and pull-ups Calf raises	Cardiovascular endurance↑ Muscle strength↑ Stretch and calf muscle strength↑	[21]: A, [27]: A, [28]: A, [29]: B, [30]: B, [31]: A
Weak muscles	Lunges and bicep curls using dumbbells Weight training High-intensity interval training Functional training, Eccentric training, Interval training Sprinting training Mid back squeezes Planks Squats Burpees Pendulum lunges	Muscle strength↑ Muscle resistance↑ Muscle strength↑ Muscle strength↑ Firm muscle↑ Back muscle strength↑ Core muscle strength↑ Core and thigh muscle strength↑ Core and whole body muscle strength↑ Core and thigh muscle strength↑	[32]: D, [33]: C, [34]: A, [35]: B
Imbalance	Standing on one foot, BOSU ball ^3^ movements Brisk walking Tai Chi Dance Cycling	Balance↑ Walking speed and steps↑ Weight shifting ability↑ Movement control↑ Trunk rotation ability↑ Multidirectional stepping↑ Maintenance of postures↑ Dynamic balance↑ Backward walking↑ Turning and Multitasking skills↑ Walking speed and steps↑	[20]: A, [22]: C, [23]: C, [36]: D
Rigidity	The shimmy ^4^ Bridge press Easy standing body transition exercise Yoga Rotation Whole body stretching Snow angels Hamstring stretch Hip flexor stretch Calf stretch Shoulder shrugs Chin tucks	Coordination and relaxation↑ Core strength and body strength↑ Body balance and coordination↑ Stretching↑ Body balance and coordination↑ Coordination↑ Stretching↑ Muscle relaxation↑ Stretching↑ Stretching↑ Stretching↑ Stretching↑ Neck stiffness improvement↑	[27]: A, [37]: A, [38]: A

^1^ “↑” represents increase; “↓” represents decrease. ^2^ The level of evidence of each reference is graded as A: overwhelming data from randomized controlled trials (RCTs); B: quasi-experimental design; C: results stem from uncontrolled, nonrandomized, and/or observational studies; D: review or evidence insufficient for categories A to C. ^3^ BOSU stands for Bionic Oscillatory Stabilization Unit, which is an inflated rubber hemisphere device using for balance training. ^4^ A dance move that requires shoulders quickly alternated back and forth with body holding still.

**Table 3 ijerph-17-02894-t003:** Exercise mode and improvement for non-motor features (except cognitive dysfunction) of Parkinson’s disease.

Symptom	Exercise Mode	Improvement	References: Grade ^4^
Constipation	Physical therapy incorporating abdominal massage ^1^	Abdominal movement range↑ ^2^	[42]: D, [43]: B, [44]: C
Depression	Tai Chi, Yoga	Flexibility↑ ^3^	[27]: A, [45]: A, [46]: A, [47]: B
Drooling	Expiratory muscle strength training	Muscle strength↑	[32]: A, [48]: A, [49]: D
Fatigue	Tango, High-intensity exercise training	Blood circulation↑ Metabolism↑	[50]: A, [51]: D, [52]: B
Orthostatic hypotension	Leg-holding exercises	Circulation↑	[33]: C, [42]: D
pain	Tai Chi, Breathing exercises, Yoga	Flexibility↑ Resistance↑ Balance↑	[27]: A, [37]: A, [38]: A, [45]: A, [46]: A, [53]: D, [54]: D
Psychosis (hallucinations or delusions)	Aerobic exercise, Yoga	Focus↑ Cardiovascular↑ Muscle strength↑ Body coordination↑	[33]: C, [34]: A, [35]: B
Sexual dysfunction	Qigong, Cycle ergometer exercise training	Focus↑ Depression symptoms↓	[46]: A, [55]: A, [56]: D
Anorexia, nausea, vomiting	General exercise	Neural apoptosis↓ Neurodegeneration process↓ PD development and symptoms↓	[57]: D, [58]: D, [59]: D
Sleep problems	Nordic Walking Aerobic training Weight lifting exercise Yoga	Muscle strength↑ Coordination↑ Physical capacity↑ Cardiorespiratory fitness↑	[28]: A, [31]: A, [42]: D, [60]: C, [61]: B, [62]: A, [63]: A, [64]: A, [65]: A
Urinary problems	Exercise-based, biofeedback-assisted behavioral intervention Pelvic floor muscle exercise	Muscle strength↑	[29]: B, [30]: B, [66]: A

^1^ To use massage techniques for pwPD with constipation problems. ^2^ “↑” represents increase; “↓” represents decrease. ^3^ Studies have shown that yoga can be considered as a therapy in treating depressive and anxiety disorders. Following along from the suggested exercise mode that targets the alleviation of the symptoms pwPD have, physical improvements then appear. ^4^ The level of evidence of each reference is graded as A: overwhelming data from randomized controlled trials (RCTs); B: quasi-experimental design; C: results stem from uncontrolled, nonrandomized, and/or observational studies; D: review or evidence insufficient for categories A to C.

**Table 4 ijerph-17-02894-t004:** Exercise modes and improvements for cognitive dysfunction.

Cognitive Domains	Exercise Mode	Improvement	References: Grade ^3^
Attention	Dual-task walking, agility course Shifting focus between tasks Walking Visual/auditory cue training Agility training	Divided ability↑ ^1^ Switching and shifting task ability↑ Selective ability↑ Alerting attention↑ Sustained↑	[25]: D, [26]: A, [68]: B
Executive function	Go/no-go punch training Muay-Thai Stretch training Strength training Balance training Endurance training Stroop walking ^2^ Visual/auditory cue training Regular exercise and performance-based physical function training	Execution↑ Inhibition↑ Updating ability↑ Muscle power↑ Balance ability↑ Muscle strength↑ Body coordination↑ Big muscle group resistance↑	[31]: D, [68]: B, [69]: A, [70]: D, [71]: D

^1^ “↑” represents increase; “↓” represents decrease. ^2^ An innovative technique that requires dual-tasking for testing executive function and detecting cognitive impairment. ^3^ The level of evidence of each reference is graded as A: overwhelming data from randomized controlled trials (RCTs); B: quasi-experimental design; C: results stem from uncontrolled, nonrandomized, and/or observational studies; D: review or evidence insufficient for categories A to C.

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
