# Peer review of "Evidence Supports PA Prescription for Parkinson’s Disease: Motor Symptoms and Non-Motor Features: A Scoping Review"

_ijerph, 2020, doi:10.3390/ijerph17082894_

Round 1

Reviewer 1 Report

The title is misleading because specific exercise prescription for PDD was not addressed. I think the title ought to be changed to something like “Evidence Supports PA prescription for PD: Motor symptoms and non-motor features”.

Use the term non-motor features, rather than non-motor symptoms.

Throughout the manuscript: Using the term Parkinson’s disease, the word “Disease” is not usually capitalized.

Many authoritative statements are made without primary citations. Please cite primary studies, trials using a randomized controlled trial design throughout the manuscript, where possible, or meta-analyses.  E.g., page 6 of 14 Line 147 states “Eventhough recent systematic reviews could not gain concrete evidence supporting efficacy of physiotherapy in PD due to methodological problems in some studies, reviews did state that additional cueing techniques could improve the efficacy of physiotherapy.” What is the source for this statement?

Provide a citation for the quote by Plato

Page 4 of 14,line 108 to 130. Please consult a recent exercise physiology text for example by the American College of Sports Medicine to define terms such as Aerobic Exercise and Anaerobic Exercise, “balance” and flexibility.

Line 120: awkward wording: “Flexibility exercise allow people to more easily do activities that require greater flexibility.” Change to something like, the purpose of flexibility exercises is to increase joint range of motion..

Table 2, citations for each of the exercise modes listed.

Why is cycling in the anaerobic group?

What is meant by the term “Firm…tone muscles”, “body circulation”, “stroop walking”, “The shimmy”?

“Physical therapy incorporating abdominal massage”?

Figure 1, page 5 of 14 depics falls as occurring in the late stages of the diusese. This is not accurate. Falls occur early and throughout the disease progress. Line 131, what is the rationale for an integrated training in PD?

Table 3, Table 4 and Table 5 are novel. Please add a column on the right hand side and add the level of evidence of each study cited as supporting evidence, e.g., RCT, quasi-experimental design, etc.

Reviewer 2 Report

In their manuscript, Exercise Prescription for the Patients with Parkinson’s disease and Parkinson’s disease dementia: A scoping review, Cheng and Su reviewed previous literature on how physical activity is associated with motor, non-motor and cognitive symptoms. Their main conclusion is that physical activity has extensive influence on the wellbeing of the patients. Overall, the manuscript is interesting and gathers recent advances in the field. Below are my detailed comments.

  • Introduction, page 1: it is unclear why the authors give an example of Parkinson disease in Taiwan.
  • The authors mention the drug levodopa. Could the authors clarify to which symptoms it is meant to etc.
  • At the end of the introduction section, the authors should clarify what will be covered in the review. Are they going to include all symptoms, or just some etc, and what is the overall goal of the review?
  • The authors should rethink the subheadings of the review to be more descriptive of what each section includes. Also, adding a few extra subheadings might be clarifying. For example, in the section 2.2 Balance Training, the authors also talk about cognitive dysfunction and freezing of gait. Please reorganize the text and sections more logical and clearer.
  • Page 3, first chapter. Could the authors elaborate why studies suggest that patients with PD are likely to reduce their levels of physical activity. Are there any reasons for this?
  • The authors might want to clarify their tables. Currently, it’s very hard to see, for example, to which exercise classification each exercise mode belongs etc.
  • Is there are need for Table 2? As this table only covers how physical activity improves the wellbeing of healthy people and not patients with parkinson’s disease, it doesn’t really seem necessary.
  • Page 4 and 5: The authors refer to “This study”. What study do they mean?
  • Figure 1, caption needs more explanation.
  • Figure 1, y-axis: Digree -> Degree
  • Figure 1, y-axis: this needs clarification. Do the symptoms increase both up and down?
  • In figure 1 the authors have made a distinction between different stages of the diseases. Could similar distinction be added to Tables 3, 4 and 5? For example, an extra column describing at what stage which exercise type would be most beneficial. Also, could the authors make any conclusions about this, i.e., how the exercise needs to change according to the disease stage?
  • I’m not sure do I understand Table 4. For example, symptom depression is associated with increase in flexibility. How does increase in flexibility improve depression? Without any clear association this seems confusing. Same for some of the other symptoms.
  • Are there any associations with physical activity and adverse effects? For example, if wrong type of exercise is practiced at a certain disease stage?
  • Section 2.2 is quite confusing, and authors should make it more logical. For example, first the authors should speak about biological changes and then symptoms and not mix them. Also, if the authors review the biological background of certain symptoms, this should be done to all symptoms.
  • Page 9. Please define abbreviation LSVT BIG and FoG.
  • The manuscript requires thorough language by a native speaker.

Round 2

Reviewer 1 Report

Thank you for addressing my concerns. 

  1. Please give deep thought to the title of your paper. I am asking you to do this because your revised paper does not address the evidence supporting PA for people living with Parkinson disease dementia at all. Yet, the title of your paper includes the PDD term.
  2. Use person first language throughout the manuscript. Instead of using the term PD patients which is only to be used to refer to people who are in-hospital patients , use of the phrase "people living with PD (pwPD) is preferred.
  3. What is the scientific basis for recommending cardiovascular exercise only in the early stage of PD and not throughout the disease course? As shown in the figure on PA and motor symptoms and Non motor features. Surely there are trials that have found benefits of aerobic exercuse (e.g. for example, cycling) effect on UPDRS, see the recent trial by Bloem et al in Lancet Neurology. And the work by Margaret Schenkman, as 2 examples.

Reviewer 2 Report

No further comments.

Author Response

Thank you very much for spending time and effort for the review. We sincerely appreciate it.

Round 3

Reviewer 1 Report

Thank you for addressing my concerns